# The Bathy-Drone: An Autonomous Uncrewed Drone-Tethered Sonar System

Antonio L. Diaz [1,*], Andrew E. Ortega [2], Henry Tingle [1], Andres Pulido [1], Orlando Cordero [2], Marisa Nelson [1], Nicholas E. Cocoves [1], Jaejeong Shin [1], Raymond R. Carthy [3], Benjamin E. Wilkinson [2] and Peter G. Ifju [1]

[1] Mechanical & Aerospace Engineering, University of Florida, Gainesville, FL 32611, USA
[2] Geomatics Program, University of Florida, Gainesville, FL 32611, USA
[3] U.S. Geological Survey, Florida Cooperative Fish and Wildlife Research Unit, University of Florida, Gainesville, FL 32611, USA
[*] Correspondence: tony52892@ufl.edu; Tel.: +1-352-294-2829

**Abstract:** A unique drone-based system for underwater mapping (bathymetry) was developed at the University of Florida. The system, called the "Bathy-drone", comprises a drone that drags, via a tether, a small vessel on the water surface in a raster pattern. The vessel is equipped with a recreational commercial off-the-shelf (COTS) sonar unit that has down-scan, side-scan, and chirp capabilities and logs GPS-referenced sonar data onboard or transmitted in real time with a telemetry link. Data can then be retrieved post mission and plotted in various ways. The system provides both isobaths and contours of bottom hardness. Extensive testing of the system was conducted on a 5 acre pond located at the University of Florida Plant Science and Education Unit in Citra, FL. Prior to performing scans of the pond, ground-truth data were acquired with an RTK GNSS unit on a pole to precisely measure the location of the bottom at over 300 locations. An assessment of the accuracy and resolution of the system was performed by comparison to the ground-truth data. The pond ground truth had an average depth of 2.30 m while the Bathy-drone measured an average 21.6 cm deeper than the ground truth, repeatable to within 2.6 cm. The results justify integration of RTK and IMU corrections. During testing, it was found that there are numerous advantages of the Bathy-drone system compared to conventional methods including ease of implementation and the ability to initiate surveys from the land by flying the system to the water or placing the platform in the water. The system is also inexpensive, lightweight, and low-volume, thus making transport convenient. The Bathy-drone can collect data at speeds of 0–24 km/h (0–15 mph) and, thus, can be used in waters with swift currents. Additionally, there are no propellers or control surfaces underwater; hence, the vessel does not tend to snag on floating vegetation and can be dragged over sandbars. An area of more than 10 acres was surveyed using the Bathy-drone in one battery charge and in less than 25 min.

**Keywords:** bathymetry; uncrewed aircraft; sonar; tethered; retention pond; hydrology; survey; recreational sonar; drone

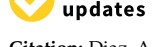



## 1. Introduction

This paper introduces a unique drone-based method to perform bathymetry on relatively small waterways with high spatial and temporal resolution. There is a wide variety of methods employed to perform bathymetry for an innumerable variety of scales and applications. Satellites can provide large-scale surveys of large bodies of water [1–3], such as lakes, bays, gulfs, and oceans, while drones and small uncrewed watercraft are increasingly used for smaller waterbodies such as rivers, inlets, retention ponds, boat basins, shipping channels, and nearshore applications [4–6]. The benefit of increased field operator safety, reduced fatigue and environmental exposure, and more accurate raster patterns are primary motivations behind uncrewed systems. The literature on drone-based and uncrewed watercraft-based bathymetry shows that a variety of sensors have been utilized,

each having their advantages and limitations. They can be grouped into airborne methods, such as photogrammetry, lidar, radar, and fluid lensing, and immersed methods such as sonar and underwater photogrammetry. Uncrewed surface vehicles (USVs) and uncrewed underwater vehicles (UUVs) allow for sensors, such as acoustic sensors, since these vehicles are in contact with the water, while aerial drones can only utilize sensors that operate with a standoff from the water surface. Typically, USVs and UUVs are slower-moving with a smaller sensor swath than aerial drones and, thus, limit the area covered during operation. In the following paragraphs, we review typical sensors, platforms, and applications associated with small UAS and uncrewed watercraft, providing the motivation and background for the development of the Bathy-drone.

Remote sensing instruments used for uncrewed bathymetry are grouped into optical or acoustic sensing which can each be accomplished with passive or active methods. Optical sensors can actively measure reflected energy, such as with immersed range-gated camera systems [7] or water-penetrating green lidar, which can reach depths of 40–50 m in clear waters [8,9]. Passively sensing reflected or scattered light is accomplished with hyper- or multispectral imaging [2,10,11] or strictly in the visible light spectrum [12], and the imagery may be georeferenced with structure from motion (SfM) [13–18] and photogrammetry [19].

Acoustic sensors typically rely on active sensing by emitting acoustic waves and measuring the reflected, scattered, and absorbed energy. Acoustic methods used in uncrewed bathymetry include sonar technologies such as multibeam and single-beam echo sounders (MBES or SBES respectively), side-scan sonars (SSS), and phase-measuring side-scan sonar (PMSS) [20]. Water-penetrating radar (WPR) [21–24] and Doppler velocity logger (DVL) or acoustic Doppler current profilers (ADCP) [25] are also active acoustic sensors that have been used in uncrewed bathymetry. In the oil and gas industry, seismographic or sub-bottom profiler (SBP) [26] sensing, either active or passive, is also a popular method of mapping.

Ancillary sensors are crucial to both optical and acoustic techniques and typically correct for positioning, heading, attitude, tide, and sound velocity. Respectively, these corrections can be accomplished with Global Navigation Satellite Systems (GNSS) antenna, gyrocompass, Inertial Measurement Unit (IMU) or Inertial Navigation System (INS), marigraph, and sound velocity profiler.

The desired data characteristics depend on the application, and uncrewed solutions are still developing. For example, deep-sea exploration can involve crewed expeditions focused on extensive coverage to capture geological features within a resolution of 50 m. Navigable coastal and inland waterways, which typically are shallow waters, have been surveyed by multiple uncrewed bathymetric platforms to meet the required rigorous navigational mapping safety standards set by agencies such as the International Hydrographic Organization (IHO) or the US Army Corps of Engineers, which can require a resolution to detect features of 0.5 m and accuracy of below 1 m uncertainty depending on under-keel clearance [27,28]. Categorizing habitats and infrastructure is another application of uncrewed bathymetric platforms and demands understanding the bottom composition of the observed area. Under any of a vast variety of scenarios, the specific application requirements guide the selection of appropriate platforms and sensors.

Bathymetry systems can be categorized as sensing-immersed such as USV and UUV or above-the-water such as satellites and UAS. Satellite altimetry provides entire models of the Earth based on radar readings of water height and slope, induced by local gravity of subsea geological features, and it is independent of water clarity with spatial resolution varying between 1 and 12 km [2,3]. Satellites such as the Hyperion hyperspectral sensor onboard NASA's EO-1 platform have collected bathymetry above coastal waters in large swaths of ~7 km to a spatial resolution of 30 m and 1–20 m of water depth depending on clarity [2]. The large swath coverage of satellites is excellent for capturing expansive geological features, but the resolution is not useful for safe maritime navigation. Satellites have also used camera sensors for depth estimation methods through wave celerity inversion, leading to sub-meter resolution, in less than 35 m clear water [29]. Fluid lensing photogrammetry

from UAS also uses camera sensors and can result in up to centimeter-scale resolution, as the technique filters for advantageous images with magnifying wave conditions [30]. The trend seen in photogrammetry is also observed with UAS radar altimetry [18] compared to satellite altimetry: a shorter sensor to area of interest distance typically leads to refined accuracy, precision, and resolution while sacrificing swath coverage. Lidar bathymetry must strike a balance, and, when paired with UAS such as Reigl's bathycopter [31], an observation height of 500–600 m can lead to accuracy of 3–5 cm at depths of 0–4 m in small to medium rivers of width 5–25 m. This tradeoff provides each optical sensor and platform pairing with unique utility, but they are all challenged by obstructions such as grassy bottom composition or flotsam, suspended particles impacting water clarity, water refraction (when not immersed) [32], and overhanging structures such as foliage or cliffs.

Acoustic sensors have been paired with USV, UUV, and UAS to great success in research and from commercial suppliers. USV with SBES, MBES, SSS, PMBS, and ADCP have all led to successful bathymetric surveys. Most acoustic sensors are deployed on UUV or USV and do not need to account for water surface reflection or refraction. Depending on the sensor and frequencies used, acoustic sensors on USV can typically record depths of 0.2 m to thousands of meters regardless of turbidity. Centimeter resolution can be achieved with sonar on USV but varies due to factors such as sensor frequency and depth. There are a variety of USV hull shapes [5] to accommodate different sensors and marine environments, such as small-waterplane-area twin hull (SWATH) [33,34], catamaran [35–37], and V hull. Commercial and research USV such as the Seafloor Systems Hydrone [38], Jetyak [39], and Searobotics Hycat [40] have successfully conducted bathymetric surveys for navigational purposes and characterization of the bottom environment in coastal and inland bodies of water. Observations from USV platforms are slower than airborne platforms due to the closer proximity, reducing the swath coverage area of the sensor, and the increased drag on the platform in water compared to air. USV and ROV deployment is also challenging as it typically requires a crewed vessel to arrive at the study site or a boat ramp for field operators to lift the vehicle into the water. USV and ROV are also typically electrically self-propelled and can struggle to steer in an autonomous mission safely and accurately against strong currents and winds [41].

Hybridization of platforms and sensor packages has led to new combinations of sensor fusion and enabled novel deployment opportunities and results. For example, combining aquatic and aerial sensors such as camera photogrammetry and sonar can improve the results of the independent sensors, especially in transitionary areas where one sensor has advantages over the other [42,43]. A unique and emerging platform hybrid is that of the tethered acoustic sensor to a UAS. Historically, this configuration can be seen on crewed helicopters with dipping sonar for submarine detection [44], and several towable sonar packages exist for crewed marine vessels such as for analyzing coral reefs [45]. Most UAS with tethered sonar uses small lightweight SBES with limited ancillary sensors [17,43,46–48]. Bobber-shaped casting sonar from Lowrance is typically used, which lacks ancillary sensors such as IMU but is excellent for extending the UAS range to take sparse point measurements by minimizing payload weight. The Bathy-drone proposed in this paper (Figure 1) is a novel approach to uncrewed bathymetry, configured as a tethered, hull-enclosed sonar with IMU and Real Time Kinematics Global Navigation Satellite System (RTK-GNSS) that can autonomously gather bathymetric data.

As evidenced by the literature, there is a trend of developing efficient uncrewed systems that are practical, inexpensive, and easy to deploy, while providing high spatial and temporal resolution for bathymetry on small bodies of water, such as ponds, rivers, boat basins, shipping lanes, and pre-construction and nearshore applications. The University of Florida has developed a system that incorporates a drone that drags a vessel/platform via a tether that can be equipped with a variety of sensors such as sonar or underwater cameras. It has advantages over USVs since the system can be flown to the survey location; thus, many surveys can be initiated from a land-based ground station, and no boats or boat ramps are needed if the location is within the FAA-required visual line of sight. Since the vessel has

no propulsion system (propellers), floating vegetation does not hamper operation. In the remainder of this manuscript, we describe the system design, the experimental procedure to assess the design, the results from our assessment, and a description of future work.

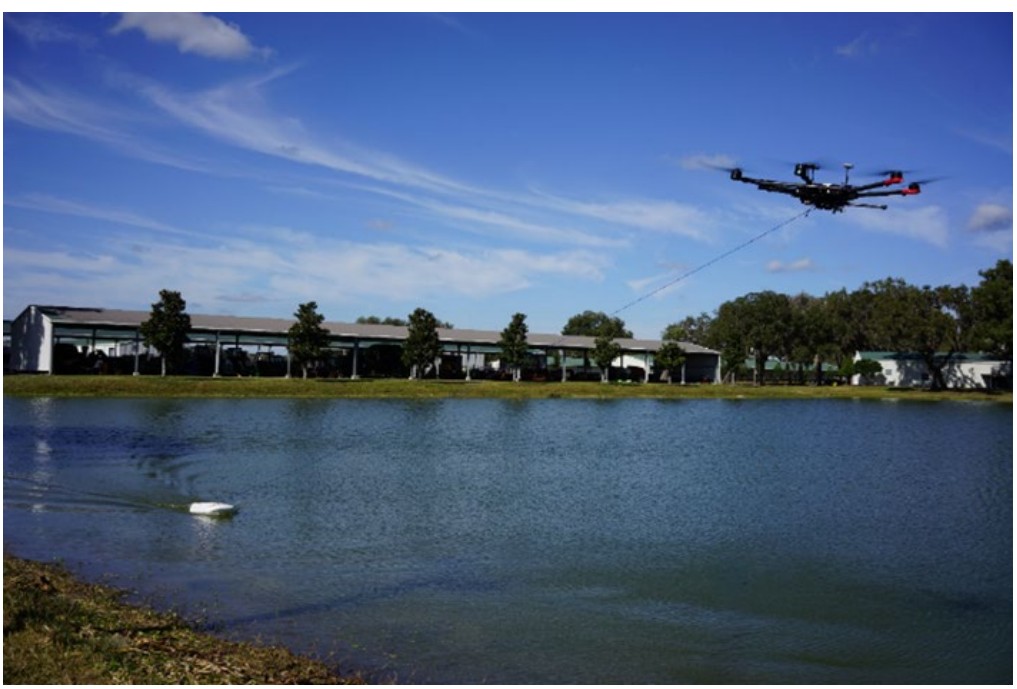

**Figure 1.** Bathymetry system during autonomous flight for ground-truthing at University of Florida Plant Science Research Citra, Florida. Photo credit: author's UF UASRP lab.

## 2. System Design

The design of the vessel for the Bathy-drone system is an important consideration for the system to perform over a wide range of speeds without capsizing in turns while maintaining a level attitude and housing the sonar components. The vessel must track in a straight line behind/below the drone to ensure that the passive vessel path is similar to the commanded drone path. Unlike most powered vessels, the driving force produced by the drone through the tether does not act in a direction parallel to the water surface. Instead, it acts upward at an angle that is determined by the length of the tether and the height of the drone above the water. Thus, the upward component of the force vector must be considered in the vessel design.

During the process of designing the hull, three basic shapes were tested, a soft-edge V-hull shape, a trimaran shape, and a skiff-like planing (which planes or skims on the water surface versus a hull that parts the water) hull. Each of these was satisfactory for straight-line portions of the mission patterns, but the lateral resistance, especially in the bow of the V-hull and trimaran, led to capsizing issues in the turns. This was documented by repeated experiments that were conducted over a wide velocity and corner radius range. The soft-edge skiff-like planing hull shape proved robust in both straight/level tracking and corner turns once a trim plate and fins were added.

The location of the tether attachment point on the vessel proved to be important to balance the forces on the hull through the entire speed range. The first attempt placed the tether attachment point on the nose of the vessel, but this led to longitudinal pitch-up as a function of speed. After analysis of the free body diagram (Figure 2) at a constant velocity, it was determined that, when attaching the tether so that the force passes through the center of gravity (CG), trim and level conditions were achieved through the entire speed range. Figure 2 shows the forces acting on the hull at a constant velocity. The tether provides a constant pulling force that is dependent on the drag force, which is horizontal, while the pulling force is angled upward. Both are dependent on the speed of the vessel/drone.

Other forces on the hull include an upward buoyancy force and a hydrodynamic force on the forward portion of the rocker line. The latter is speed-dependent and tends to produce a longitudinal pitching moment about the CG that results in the bow pitching upward as the speed increases. To compensate for this pitching moment, a trim plate was added behind the CG, angled downward, to produce a speed-dependent pitching moment in the opposite direction of the speed-dependent bow pitching moment. The trim plate is like a horizontal stabilizer on an aircraft. By placing the tether attachment location so that the force acts through the CG, and with the angle of the trim-tab empirically optimized, the vessel tracked the level through the speed range of 0 through 24 km/h. By incorporating two fins on the trim-tab, tracking of the vessel improved, in addition to allowing turns without capsizing. The fins act as a pivot point where horizontal forces through the tether swing the hull's bow around, as shown in Figure 3. The soft edges of the planing hull bow rocker provide little lateral resistance; thus, the bow swings around smoothly in the turns. This was not the case for the V-hull and trimaran shapes.

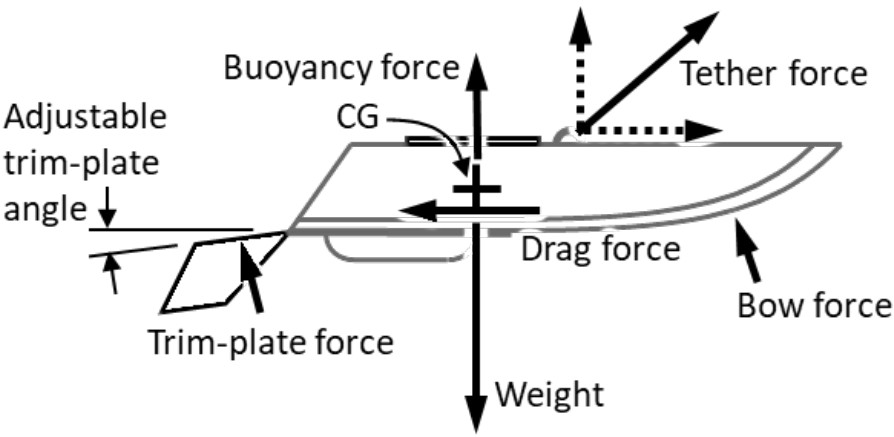

**Figure 2.** Forces at a constant speed on the hull. The center of gravity is labeled CG. Figure credit: author's UF UASRP lab.

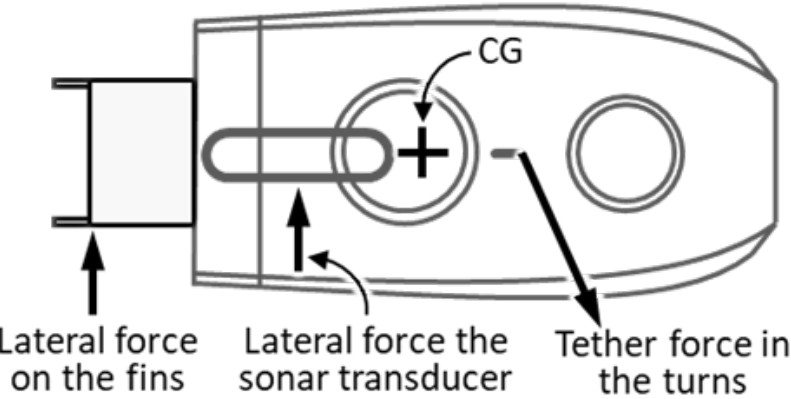

**Figure 3.** In a turn, the tether force acts laterally with respect to the hull. The center of gravity is labeled CG. Figure credit: author's UF UASRP lab.

Additional considerations in the design included providing adequate volume in the hull to house the sonar console and ports to provide easy access to the microSD cards, batteries, and console keypad. The CG was also factored into the design so that, at 0 km/h, the hull floated level. This was achieved by balancing the buoyancy force with the center of gravity of the vessel. Small adjustments were made to achieve this by slight shifts to the CG and small weight plates mounted on the top of the trim plate. The aluminum trim-plate

also acts as an electrical ground for the sonar transducer, mounted on the bottom in the aft portion of the hull.

The sonar unit is a low-cost commercial off-the-shelf (COTS) recreational fish-finder by Lowrance®, a company from Tulsa, OK, USA, model Elite ti7, with an active scan transducer. The Lowrance sonar unit features a NMEA serial data port which allows for communication between other marine electronics. By using this port, the Lowrance can output serial data packets to the ground station using an onboard telemetry link. The telemetry link is an RFD900+ long-range 900 MHz transmitter RFDesign® a company from Brisbane, Australia, that can send data up to 30 km in optimal conditions. The NMEA protocol uses a variety of 'sentences' to transmit individual data streams. The Lowrance menu allows for the selection of NMEA sentences and at what frequency they are output. The Bathy-drone currently outputs the GPS position and the sonar depth values using the following packet types: DBT—depth below transducer, DPT—depth of water, RMC—recommended minimum navigation information, GGA—global positioning system fix data, HDG—magnetic heading, deviation, variation, GLL—geographic position, latitude/longitude, and VTG—track made good and ground speed. These sentences are broadcast using the RFD900 telemetry link and are received by the ground station computer to interpret the data using the Reefmaster software package. Details of Reefmaster use are found in the ground station and data processing discussion. The sonar imagery down- and side-scan data are saved directly to the onboard microSD card to conserve telemetry bandwidth. A Pixhawk Cube orange with a Here+ RTK system was included in this design (Figure 4) for future work with RTK + IMU integration.

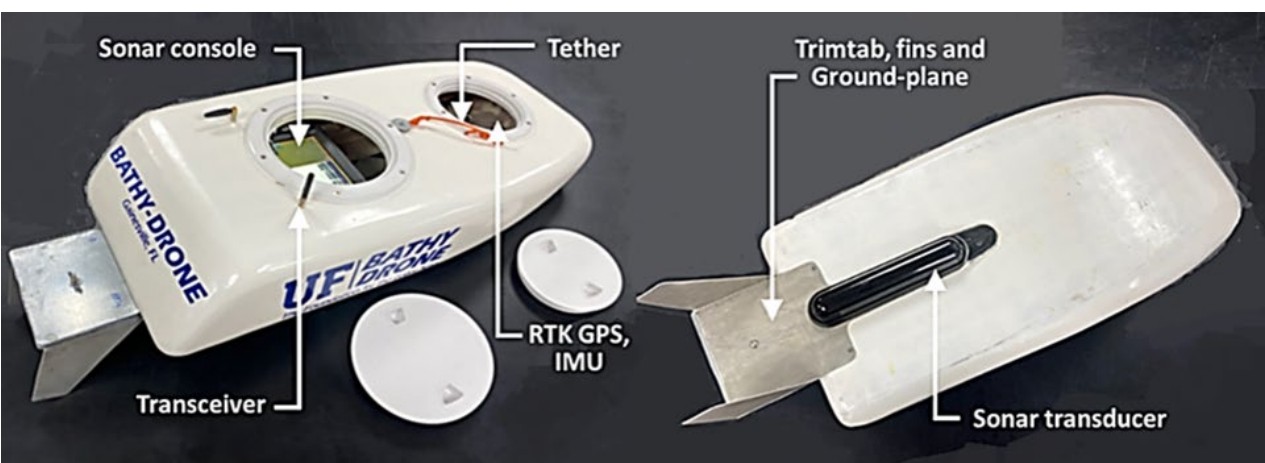

**Figure 4.** Isometric view of the Bathy-drone from the top showing sonar screen through the open hatches (**left**) and from the bottom with fins and transducer (**right**). Where RTK GPS is Real Time Kinematics corrected Global Positioning System and IMU is Inertial Measurement Unit. Photo credit: author's UF UASRP lab.

The Bathy-drone vessel is designed to be multirotor UAS agnostic if the drone can carry the weight of the vessel portion. The interface between the UAS and the vessel can be as simple as a knot. Any drone can, therefore, tow the vessel without any prior modification. The DJI® (a company from Nanshan, Shenzhen, China) Matrice 600 drone was utilized to test the Bathy-drone vessel, featuring 30 min of battery life and 6 kg payload capacity [49]. An alternate platform, the Alta X drone from Freefly Systems®, a company from Woodinville, WA, USA [50], has similar performance metrics. The current iteration of the Bathy-drone weighs just under 14 lbs and is easily airlifted by the Matrice 600 to be shuttled to the water surface.

### 3. Experimental Procedures

Traditional methods of pond survey included a field crew operating a small watercraft to measure the pond bottom in a systematic grid pattern. Measurements were taken via a level pole, range pole, or sounding wire and were interpolated to create an estimated surface of the pond bottom (Figures 5 and 6) [42,51].

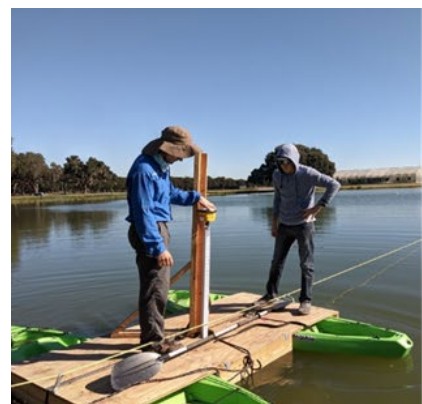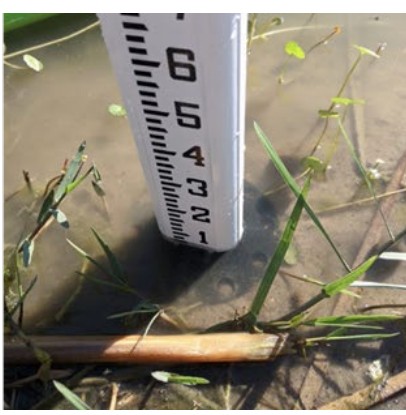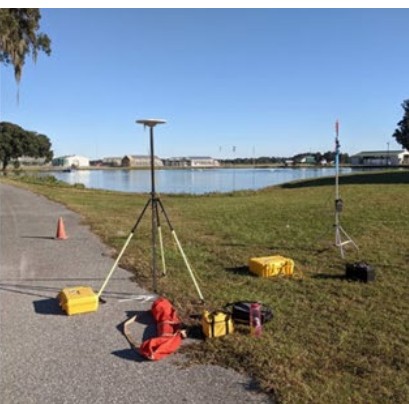

**Figure 5.** The moving rover and the data collector taking topographic points throughout the pond from a barge (**left** image). Custom 3D printed silt foot for level rod (**center**). The base station set up on a nail driven into the asphalt (**left** tripod) and the radio (**right** tripod) (**right** image) transmitting the RTK GPS corrections. The base nail was measured for 8 h as static observations and submitted to NOAA OPUS. Topographic points collected across different days were translated so that the base station points aligned with the NOAA OPUS solution. Photo credit: UF UASRP; student researchers pictured on the left image.

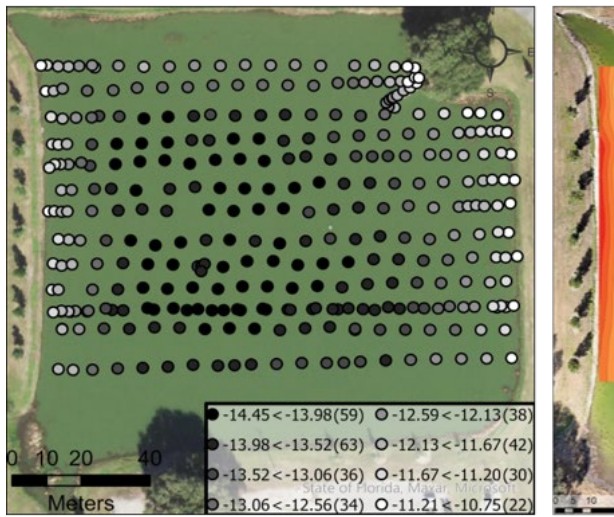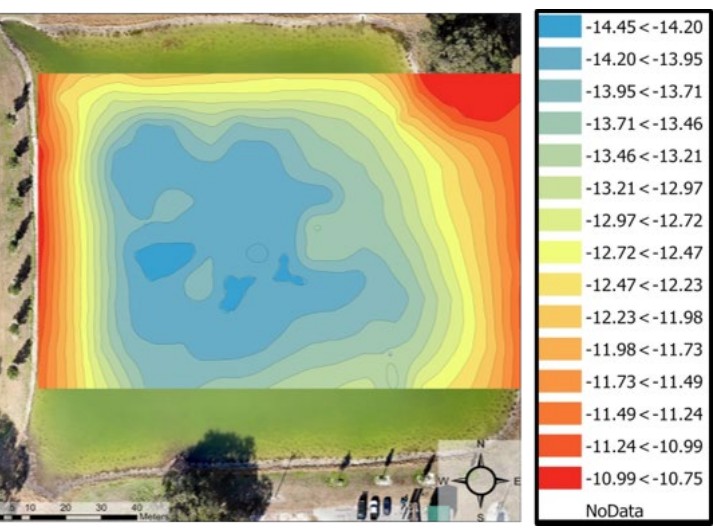

**Figure 6.** Ground-truthing data consisting of 324 RTK-corrected points with a minimum depth of 0.17 m, maximum of 3.87 m, mean of 2.30 m, and standard deviation of 1.03 m (**left**). Local polynomial interpolation of the ground-truth data (**right**). Horizontal coordinates are in NAD 1983 (2011) State Plane Florida West FIPS 0902 (meters) and vertical coordinates are in height above ellipsoid (meters).

The University of Florida Plant Science Research and Education Unit (UF PSREU) is a 1086 acre facility which conducts specialty crop research. The UF Unmanned Aircraft Systems Research Program (UASRP) uses this expansive facility to test uncrewed aircraft systems in an unpopulated area. On this property, there is a 5 acre retention pond in which we tested the bathy-drone system. Measurements from a graduated level rod were used to measure the bottom surface of the pond and compare the obtained sonar data to traditional

survey methodology (Figure 5, center image). To take direct measurements, a Trimble® (a company from Sunnyvale, CA, USA) RTK GNSS survey system was attached to the top of the level rod. RTK GNSS uses a stationary base station to transmit corrections to a moving rover with a radio or internet data link (right image). These computed corrections allow the rover to obtain centimeter-level accuracy point measurements. Data from the base station were entered into NOAA OPUS [52] to translate the observations gathered into the national spatial reference system (Figure 6). The ground-truthing experiments used the Trimble SPS855 as the base station receiver and the SPS986 as the rover platform. Corrections were transmitted using a high-power 35 W TDL450 radio. The level rod and observation barge were tethered to a rope that was strung across the length of the pond (Figure 5, left image). The rope was graduated with markings every 7.6 m (25 feet). Each of the points surveyed for the ground-truthing had a report of associated observation notes and statistics such as dilution of precision, number of satellites, horizontal precision (DRMS), vertical precision (1 sigma), and tilt distance. As a summary of the measurement uncertainty associated with the ground-truthing points gathered, the reported tilt distance average was 6 cm with a standard deviation of 5 cm, horizontal precision average was 1 cm with a standard deviation of 1 cm, and the vertical precision average was 2 cm with a standard deviation of 1 cm.

The file containing the waypoints of an autonomous flight mission was created in an external software package such as ArcGIS® Pro a company from Redlands, CA, USA or Google Earth® a company from Mountain View, CA, USA. The ground-truthing mission used ArcGIS® Pro point creation tools to create equally spaced waypoints and the line creation tool for the flight lines. The "export to KMZ" geoprocessing tool was then used to create a file compatible with DJI®, China, ground station pro software and with the DJI® Matrice 600 multirotor drone. Ground Station Pro allows operators to set mission parameters such as speed and elevation and transmits the preplanned mission to the drone. The pilot can see the mission status, as well as battery and GPS signal levels, live throughout the flight mission. The autonomous turn settings are changed from stop and turn to continuous radius turns. Radius turns prevent capsizing from abrupt slackening and tugging of the towline. The radius is programmed to be as large as possible between transects which can vary in spacing depending on the resulting resolution desired from the sonar data. Alternatively, autonomous missions can be planned on the open-source software Mission Planner.

The current flight field procedures were developed for ground-truthing of the system and gathering data for isobaths that tie to geodetic datums. The vessel electronics were turned on and set to record before takeoff of the UAS. Standard flight operating procedures and safety were followed. The pilot navigated the drone to the first waypoint lifting the vessel above any obstructions. The drone was lowered to the flight altitude of 6 m (20 feet), placing the vessel on the water. With the drone and vessel at the starting waypoint, the autonomous mission was enabled. The autopilot flew the drone and vessel to the recovery point while the pilot maintained the ability to interfere if necessary. A top-of-water surface reading was taken with RTK GNSS to offset the sonar measurements and translate the measurements to the coordinate system of the geodetic datums. Figure 7 shows a typical path for both the multirotor drone (solid red line) and the vessel (white points).

ReefMaster is a commercial software package from Australia used for the interpretation and analysis of data from recreational fish finders and sonar units. ReefMaster supports data import from Lowrance and Hummingbird brand sonar units and can export generic point data such as shapefiles and CSV files. The combination of USV with fish finder sonar data processed through Reefmaster has proven results in bathymetry, ecological classification, and bottom characterization in multiple projects [15,53].

The software acts as part of the ground station during the flight in the field by receiving data over a telemetry link. Live data visualization on the ground station is achieved by processing depth and position updates using the NMEA protocol. Live telemetry only supports the transmission of depth value data; raw sonar data must be saved to and read

from the microSD card. The Bathy-drone sonar package from Lowrance produces an sl2 file format to store the raw data from the three-in-one sonar transducer onboard. Reefmaster software can import the sl2 file from the Lowrance microSD between missions for sonar data visualization on the ground station. Rapid on-site isobaths and bottom hardness plots can be created on the ground station running Reefmaster to inform the next mission. Additionally, Reefmaster can export CSV and shapefiles for further processing on Excel, MATLAB, Python, and ArcGIS Pro.

The live link prevents loss of data during malfunctions such as power loss and adds redundancy to the onboard storage. The on-field review helps ensure that the operators met the mission requirements before returning to the office.

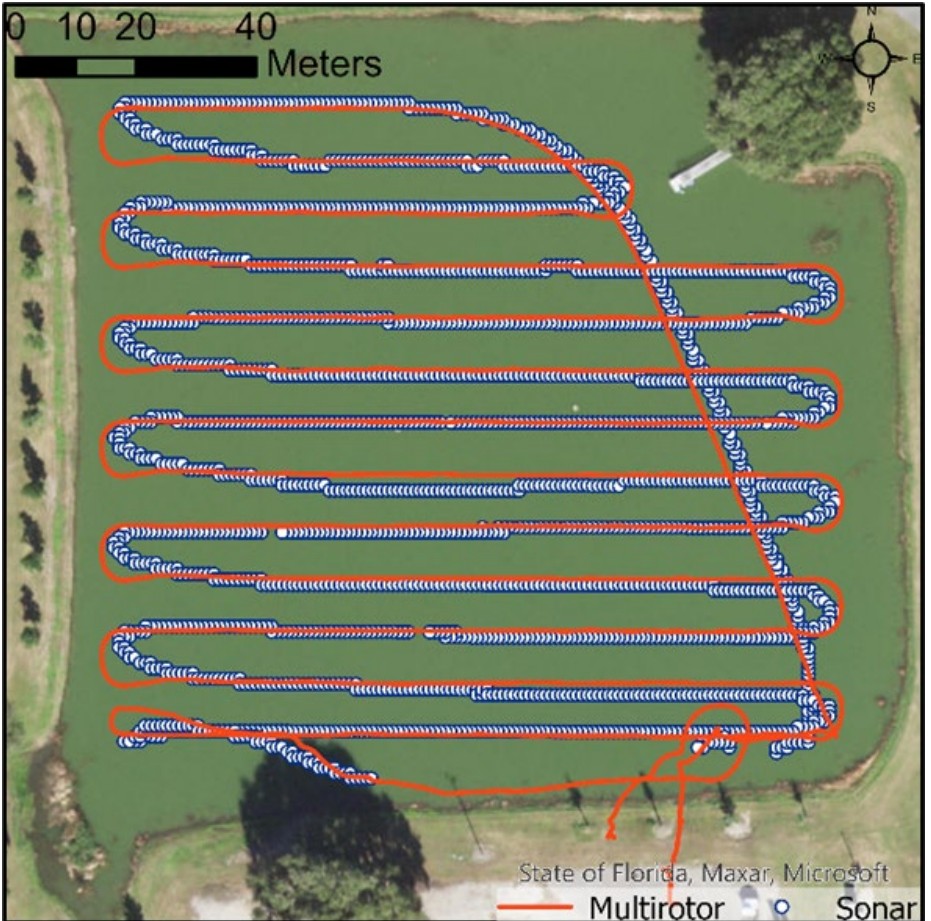

**Figure 7.** Comparison of GNSS-recorded flight path of multirotor drone and the path of the sonar payload vessel. Photo credit: UF UASRP using satellite imagery.

## 4. Results

The on-field results generated by preliminary processing of the raw sonar data on Reefmaster with further processing on ArcGIS Pro included isobaths (Figure 8), bottom hardness (Figure 9), and sonar side-scan imagery (Figure 10). According to the experimental procedures in Section 3, the primary desired outputs from the surveys included isobaths and bottom hardness contours. These data are presented here as contour plots in Figures 8 and 9.

To compare the ground-truth survey points to the sonar depth data, both datasets were imported into ArcGIS pro. The sonar data were converted to the same coordinate system as the ground-truth points, such that horizontal coordinates were in State Plane Florida West FIPS 0902 (meters) using the 2011 realization of NAD 1983 and vertical coordinates were in height above ellipsoid (meters). In addition, the sonar data must have the water level

subtracted, which is currently measured at the start of the flight using the RTK GNSS level rod. Subtracting the water surface elevation is like accounting for the length of a range pole or, in this case, the level rod, when taking rover measurements with RTK GNSS.

The ArcGIS Pro geostatistical wizard is a toolset that guides users through the process of making interpolation-based surfaces in a step-by-step manner. The geostatistical wizard uses point data to create an interpretive raster surface. To use this tool, the sonar data are imported into ArcGIS from Reefmaster as a point shapefile, with each point representing an individual sonar depth observation. The geostatistical wizard uses the point sonar data with a selected area interpolation method. Local polynomial interpolation was used to create the surface, and the extents were clipped in the processing environments tab (Figure 8). The surface interpolation was restricted to the limits of the sonar data capture to minimize extrapolation. The GA layer to points tool was then used to compare the acquired ground truth survey points to the interpolated sonar data. The GA layer to points tool is used to measure the accuracy of a predicted surface by comparing it to known point observations. In this case, accuracy was assessed by comparing the predicted surface output from the sonar to the points measured using the RTK GNSS level rod (Figures 11 and 12).

The detailed measure of accuracy indirectly captures the horizontal accuracy of the sonar data within the depth measurements. To isolate horizontal accuracy from vertical accuracy, the quoted accuracy of the GNSS devices used can be considered. The Trimble RTK GNSS equipment is positionally accurate with a horizontal accuracy of 1–2 cm, while Pixhawk Cube Orange GNSS RTK internal to the vessel can be as accurate as 3 cm, and the Lowrance Elite Ti2 claims 20 m RMS positioning accuracy [54]. A simple preliminary experiment comparing the three devices observed that the Lowrance GPS can waiver from the RTK GNSS devices by 1.5–3 m. However, the uncorrected GPS is, in principle, mostly translated horizontally and still captures the relative path taken by the vessel [55].

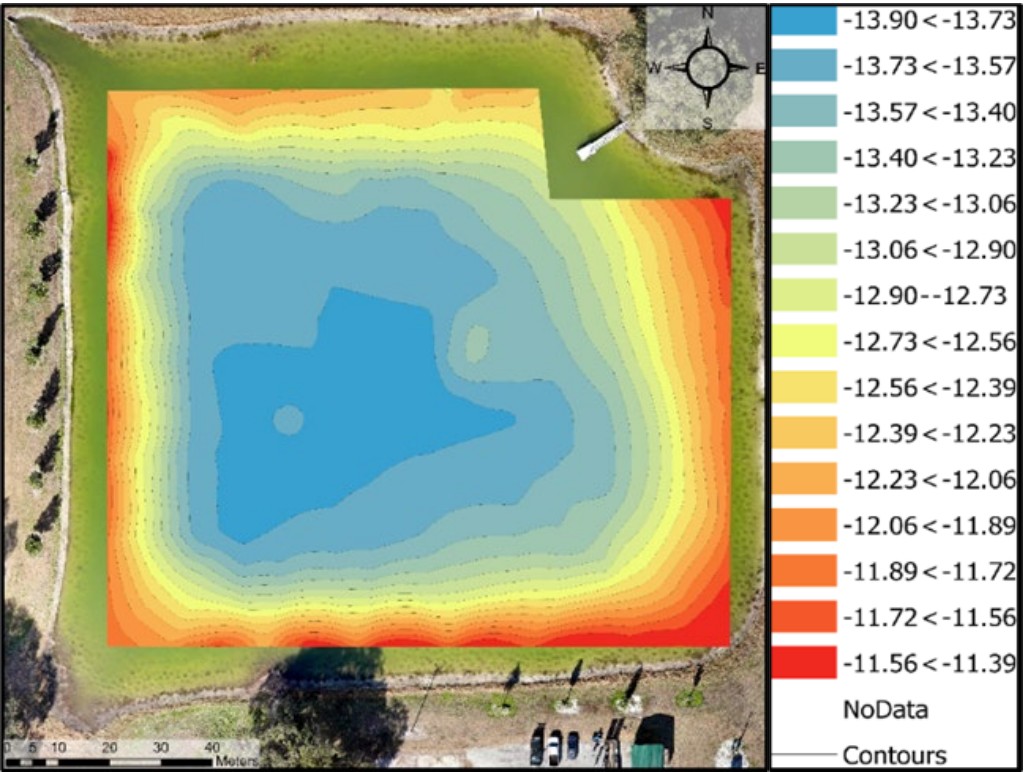

**Figure 8.** Local polynomial interpolation of sonar data from a cross boustrophedon flight pattern at 4.5 mph. Superimposed on photogrammetry data gathered by author's UF UASRP lab. Photo credit: UF UASRP.

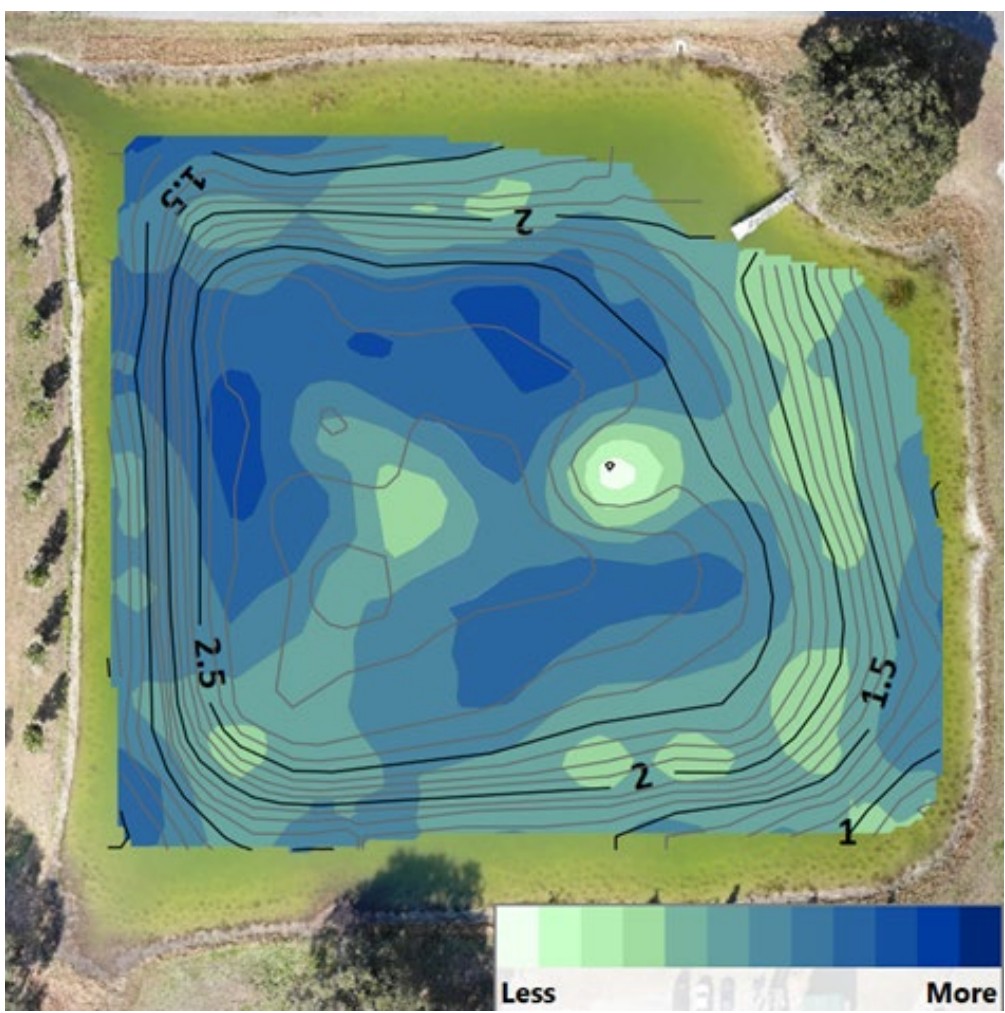

**Figure 9.** Bottom hardness as a measure of acoustic backscatter where light colors are softer and darker colors are harder. The bottom hardness color plot is overlayed with isobaths in meters. Superimposed on photogrammetry data gathered by author's UF UASRP lab.

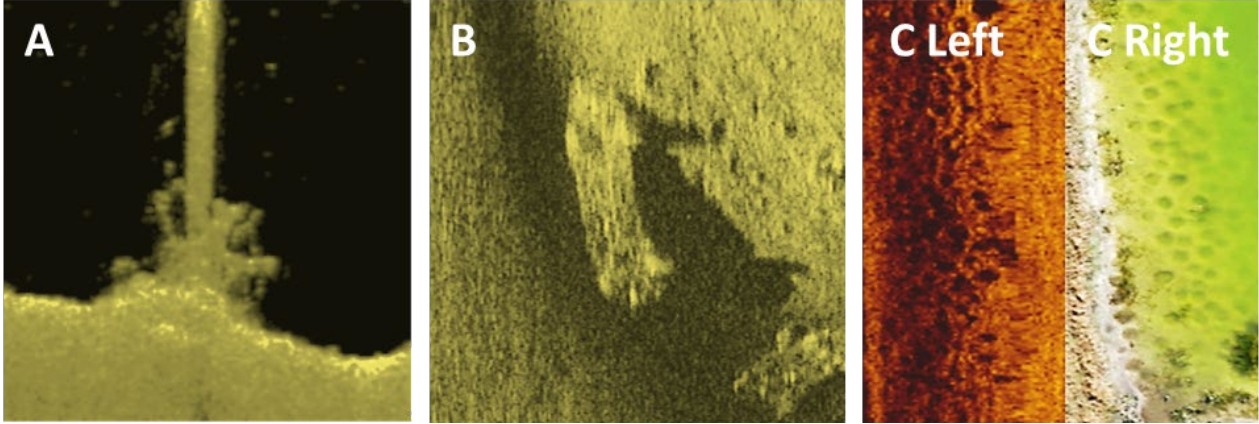

**Figure 10.** Sonar of Canal C-11, bridge pile at Fort Lauderdale Florida, showing accumulated vegetation and scour undermining the pile (**A**). This shows the potential for the drone bathymetry system to be used for inspection of civil infrastructure. Side-scan sonar of submersed vehicle in a quarry in northern Florida (**B**). ((**C**), **Left**) Sidescan sonar image of tilapia nesting beds captured at the Citra retention pond compared with photogrammetry ((**C**), **Right**). Photo credit: UF UASRP.

To compare one unique flight to another, raster surfaces were created. The GA layer to raster tool was used to create raster surfaces from each of the obtained datasets. The individual raster was compared using the raster calculator, which was used to subtract the two from each other. By subtracting the two rasters, the distribution of the difference between the two surfaces became apparent (Figure 13).

A second approach to analyze the precision of the system depth measurements is to compare the values at each intersection of the north–south (NS) and east–west (EW) raster paths, as illustrated in Figure 13 where two example intersections are highlighted with a star. The difference in the two depth values from NS and EW lawnmower paths was computed using the black intersection line shown in Figure 14. The intersecting points were found by dividing the raster pattern into straight-line segments, excluding the curved part of the trajectories, and then running an algorithm that iterated for all the NS and the EW paths to find the samples with the minimum distance between each other (ideally zero for exact intersections). The 10 closest points from the intersection points were used to fit a 10th-order polynomial of the depth values, and the precision was calculated as the difference in the polynomial depth values at the intersection points. A total of 144 intersecting points were used for this precision analysis. The result is summarized as a histogram in Figure 15, where the mean and median values of all the precision calculations were −1.37 cm and 0.89 cm respectively, and the standard deviation was 19.9 cm.

System analysis of depth reflects horizontal and vertical errors. These errors were characterized statistically from the ground-truth data and measured in multiple approaches for accuracy and precision. The accuracy measures point toward potential improvement if IMU and RTK data are used to correct the depth values. This conclusion is drawn from the sonar consistently reading deeper than the ground-truth data on average by 21.6 cm, with most of the error focused on the deeper area, as highlighted by the scatterplot (Figure 12). The deeper this system is deployed without correction, the more pronounced small-angle changes in attitude will impact accuracy. The major limiting factor of improving accuracy, to be addressed by incorporating IMU and RTK corrections, is also supported by the relative precision between independent surveys.

Ensuring the vessel is properly placed in the water at the first waypoint requires pilot skill and familiarity with the system. Improvements to this issue will be discussed as future work. The Bathy-drone at its current stage has successfully demonstrated rapid deployment in difficult-to-access waters to gather bathymetry, backscatter, and sonar imagery affordably.

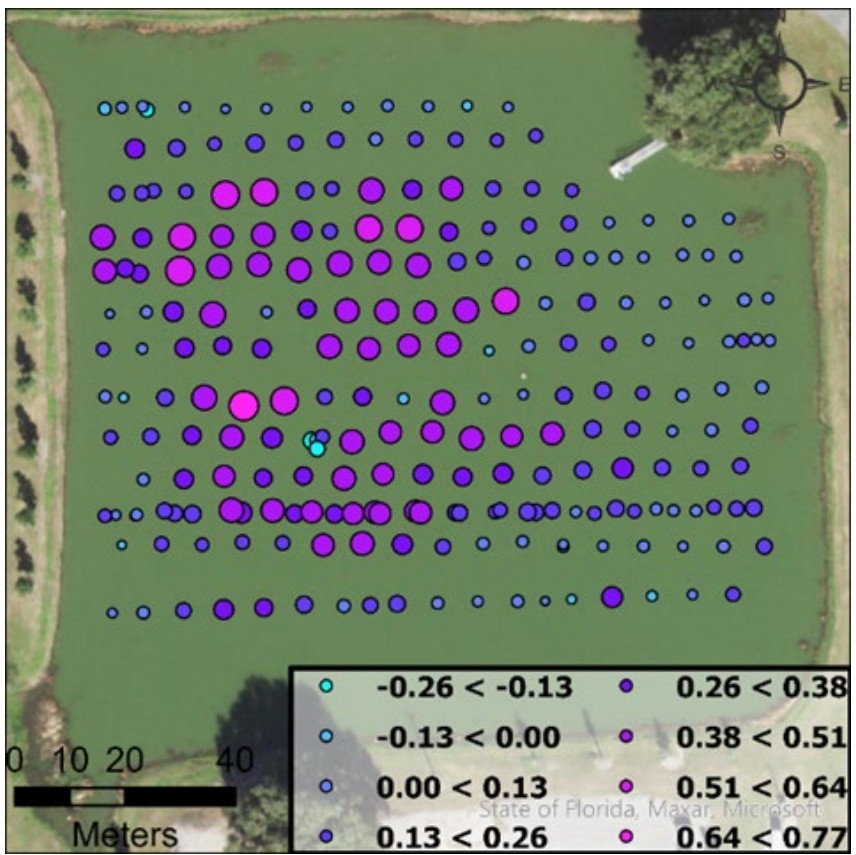

**Figure 11.** Residuals (m) between local polynomial interpolation of combined sonar transects and ground-truth data obtained from the Real Time Kinematics (RTK) corrected graduated rod measurements. Positive residuals indicate that sonar readings are deeper than ground truth. The size of the points corresponds with the residual magnitude, and the color corresponds with the direction. Photo credit: UF UASRP using satellite imagery.

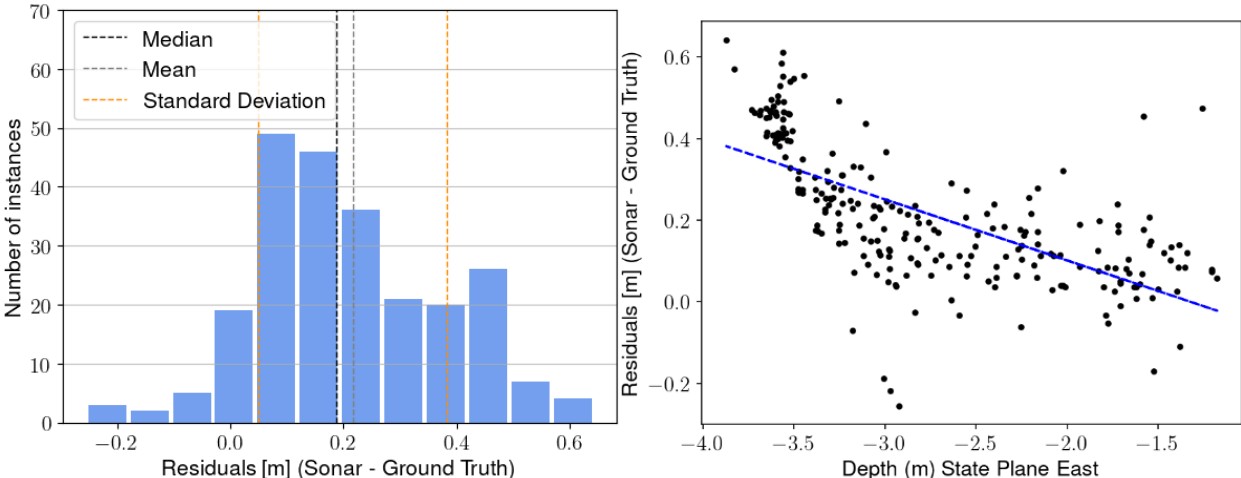

**Figure 12.** Histogram and summary statistics of residuals where the mean is 21.6 cm, the median is 18.7 cm, and the standard deviation is 16.7 cm (**left**). Scatterplot of residuals and the relationship with depth, where the blue dash linear line of best fit demonstrates decreasing residual with decreasing depth (**right**).

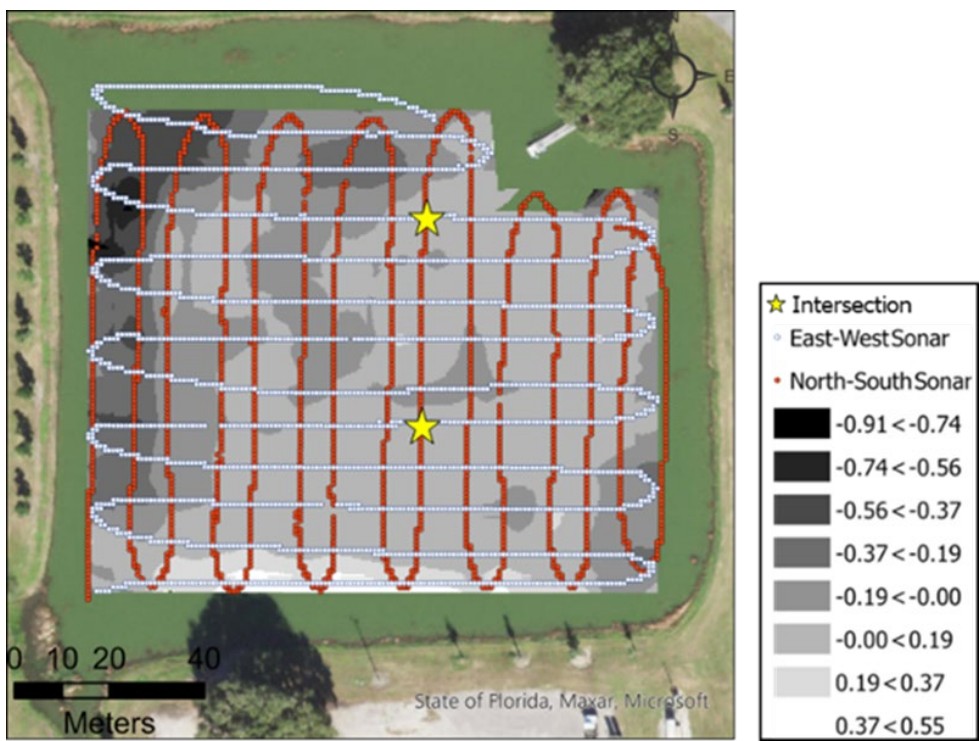

**Figure 13.** Visualization of east–west interpolated surface versus that of north–south transects. The mean residual is −2.64 cm, the median is 0.95 cm, and the standard deviation is 16.98 cm. The stars labeled "intersection" are example calculation locations for the depth difference between NS and EW. Photo credit: UF UASRP using satellite imagery.

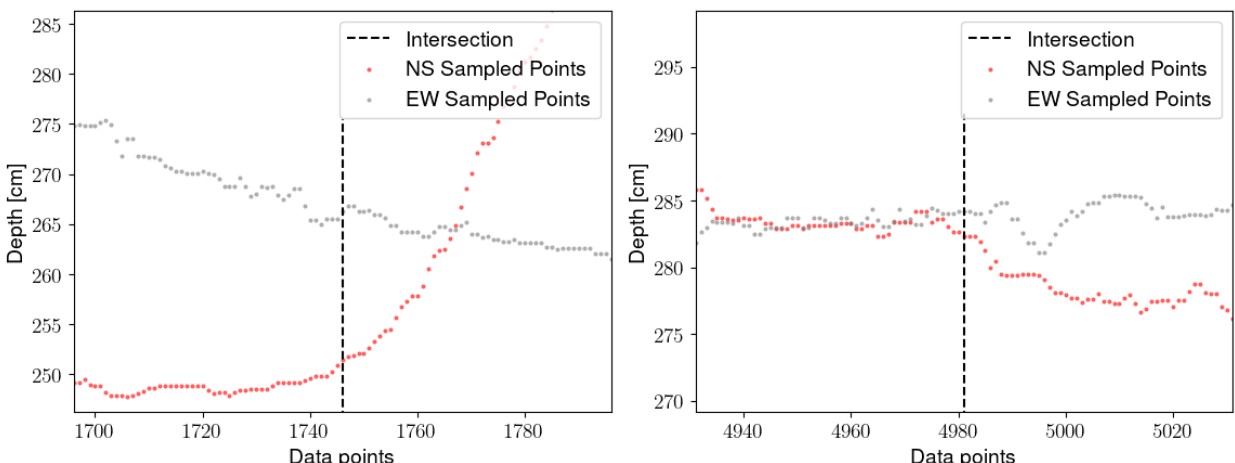

**Figure 14.** Depth values of the data values shown in red in Figure 13 on the upper star and lower star, respectively. The point where the two intersect is shown (black line). The difference for the left plot is 14.9 cm, and the difference for right plot is −1.69 cm.

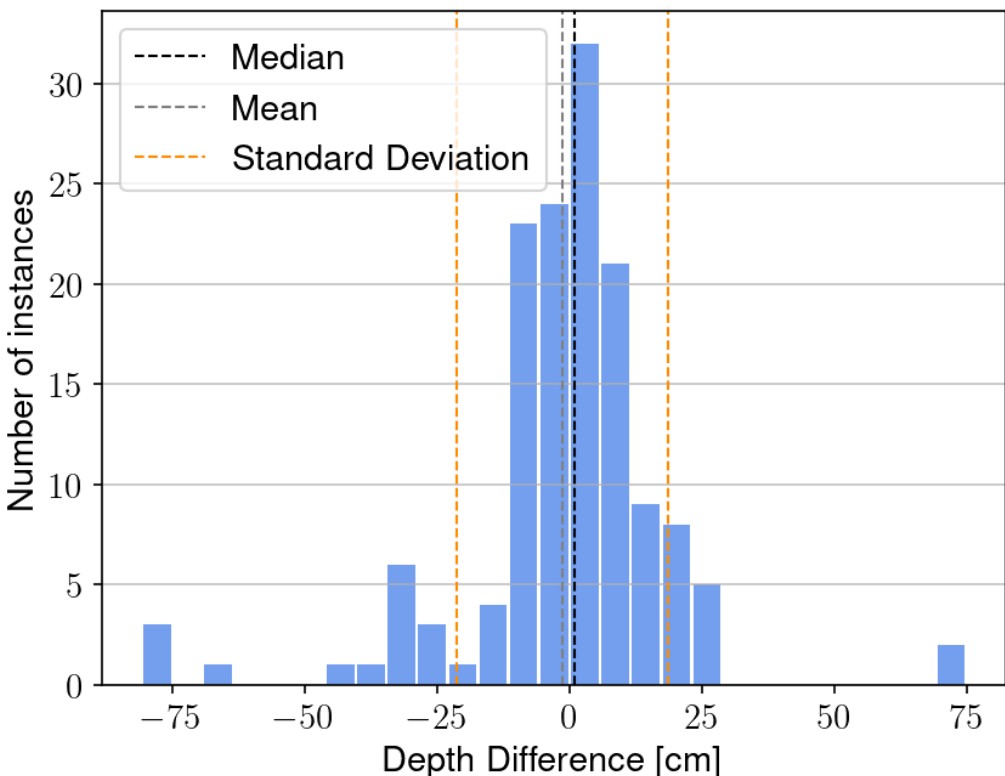

**Figure 15.** Histogram (blue) of precision analysis: difference of depth measurements at the intersections of NS and EW lawnmower paths.

## 5. Future Work

The multirotor towable bathymetric system presented in this paper results from the multiple cycles of the iterative design process. Through the design, manufacture, assembly, field testing, and data processing cycle, there were lessons learned and insights into the project's potential future direction. A literature review and collaboration with field operators in bridge infrastructure inspection and marine geomatics guided the current design and will continue to guide future additions. Currently, the bathymetric system is limited to open areas to avoid obstacles that can interfere with the safe operation of the multirotor UAS. Obstacles such as bridges, pilings, and trees are commonly near bodies of water. To avoid these hazards, active vessel control with a servo-actuated rudder would allow for autonomous control semi-independent of the multirotor UAS. Active control of the vessel necessitates that the multirotor flight altitude changes actively, an active winch mechanism, or both to maintain consistent tether tension. In either case, an emergency break-away tether connection can help reduce the chance of losing the UAS into the water if the vessel is snagged.

A new hull design inspired by field testing finished the composite layup manufacturing process. The new design intends to reduce capsizing during aggressive turns and the initial placement on the water, thereby reducing pilot fatigue and extending mission efficiency. Reduced capsizing is achieved with a rounded hull deck and low center of gravity akin to self-righting crewed vessels. A larger hatch allowing for easier access to internal payload will potentially also be tested for structural integrity and environmental intrusion. The stabilizing fins and rudder are reduced in depth to increase operational capability in shallow areas further and avoid groundings, flotsam, and floating vegetation.

The additional step of measuring the area's water level to be scanned so that the data can be tied to a coordinate reference system, which can be simplified with an onboard RTK GNSS receiver. The current bathymetric system had both the RTK GNSS receiver and the IMU installed for experimentation, but the position and attitude data would ultimately be processed onboard to correct the sonar and broadcast the result live to the ground station.

Upgrading the SBES sonar sensor could improve the swath coverage and reduce interpolation between transects. Additionally, underwater geometric features can be submerged in the pond to assess the resolution of locating and identifying structures of interest.

Currently, the sonar imagery generated from side-scan and down-scan is only used for qualitative assessment and cursory characterization of the marine ecology and environment, obstacles, and infrastructure (Figure 10). Ground-truthing the bottom hardness measures will provide more confidence in quantifying the backscatter data. Generating point clouds for three-dimensional reconstruction of the sonar data will assist in bottom visualization. The sonar imagery can also be used for object identification and avoidance or to inform further exploration of areas of interest (AOIs). Integrating communication between the drone and the vessel can allow the vessel to command the drone to explore a new AOI.

## 6. Conclusions

The uncrewed Bathy-drone is a novel configuration for rapid bathymetry and bottom characterization. The tethered vessel is towed autonomously on a preprogrammed mission by a multirotor drone. The surface vessel has a fixed recreational COTS sonar with down-scan, side-scan, and chirp capabilities gathering sonar imagery, bathymetry, and bottom hardness data. The raw field data are stored on board using a microSD card and transmitted live via telemetry link. This paper discussed state-of-the-art approaches in small, uncrewed bathymetry, the Bathy-drone design, and ground-truthing of the system against traditional RTK GNSS. An assessment of the accuracy and precision of the system based on the ground-truthing results provides insight toward future ancillary sensors. Field operation of the Bathy-drone system highlighted novel advantages unique to the tethered configuration. Surveys are based on land as the multirotor airlifts the vessel to the first waypoint on water allowing for rapid deployment and reduced crew fatigue. Boat docks are unnecessary as the Bathy-drone can fly over obstacles such as mud flats, sandbars, tree lines, and fences. Transport of the system and ground station to the area of interest via crewed boat or vehicle is convenient due to the low weight and volume. The propulsion is provided by the multirotor drone allowing the vessel to glide over swift current, debris, floating vegetation, and the ground. Speeds of 0–24 km/h (0–15 mph) were tested, and an area of more than 40,000 m$^2$ (10 acres) was surveyed with one battery charge in less than 25 min. Field testing will continue to inform new applications and iterations. Implementation of the future work discussed will continue to expand the capabilities and applications of the system.

**Author Contributions:** Conceptualization of the multirotor tethered sonar configuration, A.E.O. and P.G.I.; methodology of ground-truthing experiments, A.L.D., A.E.O., P.G.I. and B.E.W. software developed and implemented for data processing, A.L.D., A.E.O., A.P. and J.S.; validation, A.L.D., A.E.O. and A.P.; formal analysis, A.L.D., A.E.O. and A.P.; investigation, A.L.D., A.E.O., A.P., H.T., O.C., M.N. and N.E.C.; resources, P.G.I., B.E.W. and R.R.C.; data curation, A.L.D., A.E.O. and A.P.; writing—original draft preparation, A.L.D., A.E.O., P.G.I. and A.P.; writing—review and editing, A.L.D., A.E.O., A.P., B.E.W. and P.G.I.; visualization, A.L.D., A.E.O. and A.P.; supervision, P.G.I., J.S., B.E.W. and R.R.C.; project administration, P.G.I., J.S., B.E.W. and R.R.C.; funding acquisition, P.G.I. and R.R.C. All authors read and agreed to the published version of the manuscript.

**Funding:** This work was internally funded by the University of Florida.

**Conflicts of Interest:** The authors declare no conflict of interest. The funders had no role in the design of the study; in the collection, analyses, or interpretation of data; in the writing of the manuscript; or in the decision to publish the results.

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
