# Peer review of "The Bathy-Drone: An Autonomous Uncrewed Drone-Tethered Sonar System"

_drones, doi:10.3390/drones6100294_

Round 1

Reviewer 1 Report

The authors have addressed my review points sufficiently and this manuscript is now ready for publication

Reviewer 2 Report

It is an interesting approach to solving the problem of modelling the bottom of a lake, river, sea...

The article is well prepared, and the results are clearly presented here.

I don't have any fundamental comments on the article.

This manuscript is a resubmission of an earlier submission. The following is a list of the peer review reports and author responses from that submission.

Round 1

Reviewer 1 Report

This paper presents an autonomous unmanned drone-tethered sonar system. In general it reads well. However, the paper is not very specific in its message. It can be shortened at many locations. It sometimes reads as an advertisement of the system.

- the introduction presents an enormous amount of ways to determine bathymetry. This is really not needed. It also seems somewhat misleading. The ship mounted MBES and SBES are still the most common way to determine  bathymetry with the resolution typically aimed at for the application considered in this contribution. In addition, their signals can be used for classification. This paper is in line with trends to apply these type of sensors to unmanned systems.

- design of the vessel: I would think this is not needed for this paper. The topics addressed are already very scattered. Considering the focus of the journal, a focus towards the remote sensing application should be brought in the paper.

- On page 6 it is mentioned that the drone can operate for 30 minutes. This seems extremely short to me. I can imagine this affects the applications that can be considered?

- what are the methods applied to realize the results of figure 8 and 9. Why not just plot actual bathymetry instead of ranges in figure 8? How is hardness (figure 9) determined and what does it mean?

- at what locations are the results of figure 10 taken? Please add markers in figure 8 and 9.

- text on lines 400-404 on page 12 is not clear and should be improved. How are the horizontal and vertical accuracy determined? This is very important for assessing the results.

- Nothing is done with the results of figure 9. This figure can be removed. What is now exactly the cause of the 21.6 cm offset?

- page 6, line 221: what is an active scan transducer? What are the frequencies that can be emitted (line 227, page 6). What frequencies are used?

Detailed comments:

-abstract: "plotted in various ways". This can be eliminated. Nothing really stated here.

- abstract: "An assessment of the accuracy ..." -> "The accuracy ...."

- abstract: The sentence "The results justify ..." can not be justified at all from the presented information.

- page 1, line 39: "perform bathymetry"?

- page 2, line 50" "sensors" -> "underwater sensors"

- page 6, line 239: what is agnostic?

- what exactly is the ground-truthing mission mentioned on page 8, line 293? Please specify.

- text on page 8 (lines 300-304) is almost the same as that on page 9 (lines 324-329)

- text on page 8 and page 9 (but also other locations) can be significantly shortened. It contains info that is more suitable for a technical report. Same for text on pages 11 and 12.

- caption of figure 11: does ground-truth correspond to the pole measurements? Please explain. What does the size of the circles mean?  Also the color scale is not very clear. All purples are very much alike.

Reviewer 2 Report

Even submersed optical sensors need to account for refractive effects as sensors are in fact separated from water by a viewport - planar or hemispherical. Thus, light beams bend twice before being recorded by a sensor.

Figure 6. Local polynomial interpolation shows depth around 10 m where the image shows presence of vegetation on land. Is it an artifact of the chosen interpolation method?

Paragraphs starting with lines 291 and 316 are exactly the same.

Reviewer 3 Report

The authors present a novel uncrewed platform for seafloor mapping and they claim it has certain advantages over other uncrewed platforms. The study is based on well-designed experiments and results and conclusions are scientifically sound. Currently there are several USV types for seafloor mapping that require minimum effort for platform set-up and surveying. Although the study is of sufficient quality, the actual benefits of such a platform remain unclear and the presented advantages are minor compared to available alternative systems. The authors should demonstrate better the actual advantages of their platform against typical uncrewed surface vehicles (USVs) and provide specific suggestions and realistic survey examples for supporting their arguments.